# Potential of Fatty Acid Amide Hydrolase (FAAH), Monoacylglycerol Lipase (MAGL), and Diacylglycerol Lipase (DAGL) Enzymes as Targets for Obesity Treatment: A Narrative Review

**DOI:** 10.3390/ph14121316

**Published:** 2021-12-17

**Authors:** Justin Matheson, Xin Ming Matthew Zhou, Zoe Bourgault, Bernard Le Foll

**Affiliations:** 1Translational Addiction Research Laboratory, Campbell Family Mental Health Institute, Centre for Addiction and Mental Health, 33 Ursula Franklin Street, Toronto, ON M5S 2S1, Canada; matt.zhou@mail.utoronto.ca (X.M.M.Z.); zoe.bourgault@camh.ca (Z.B.); Bernard.lefoll@camh.ca (B.L.F.); 2Department of Pharmacology and Toxicology, Faculty of Medicine, University of Toronto, 27 King’s College Circle, Toronto, ON M5S 3H7, Canada; 3Addictions Division, Centre for Addiction and Mental Health, 100 Stokes Street, Toronto, ON M6J 1H4, Canada; 4Department of Psychiatry, Faculty of Medicine, University of Toronto, 250 College Street, Toronto, ON M5T 1R8, Canada; 5Institute of Medical Sciences, University of Toronto, 1 King’s College Circle, Room 2374, Toronto, ON M5S 1A8, Canada; 6Department of Family and Community Medicine, University of Toronto, 500 University Avenue, 5th Floor, Toronto, ON M5G 1V7, Canada

**Keywords:** obesity, endocannabinoid system, FAAH, MAGL, DAGL

## Abstract

The endocannabinoid system (ECS) plays an integral role in maintaining metabolic homeostasis and may affect hunger, caloric intake, and nutrient absorption. Obesity has been associated with higher levels of the endogenous cannabinoid transmitters (endocannabinoids). Therefore, the ECS is an important target in obesity treatment. Modulating the enzymes that synthesize and degrade endocannabinoids, namely fatty acid amide hydrolase (FAAH), monoacylglycerol lipase (MAGL), and diacylglycerol lipase (DAGL), may be a promising strategy to treat obesity. This review aims to synthesize all studies investigating pharmacological or genetic manipulation of FAAH, MAGL, or DAGL enzymes in association with obesity-related measures. Pharmacological inhibition or genetic deletion of FAAH tended to promote an obesogenic state in animal models, though the relationships between human *FAAH* polymorphisms and obesity-related outcomes were heterogeneous, which could be due to FAAH having both pro-appetitive and anti-appetitive substrates. Genetic deletion of *Mgll* and *Dagla* as well as pharmacological inhibition of DAGL tended to reduce body weight and improve metabolic state in animal studies, though the effects of *Mgll* manipulation were tissue-dependent. Monitoring changes in body weight in ongoing clinical trials of FAAH inhibitors may clarify whether FAAH inhibition is a potential therapeutic strategy for treatment obesity. More preclinical work is needed to characterize the role of MAGL and DAGL modulation in obesity-related outcomes.

## 1. Introduction

Obesity is a significant and ever-growing public health concern defined by a body mass index (BMI) of greater than 30 kg/m^2^. Global data indicate that age-standardized obesity rates rose from 3.2% in 1975 to 10.8% in 2014 for men and from 6.4% in 1975 to 14.9% in 2014 for women [1]. The latest data from the NCD Risk Factor Collaboration showed that in 2016, almost 2 billion adults (39% of the global adult population) were estimated to be overweight (BMI ≥ 25 kg/m^2^) and 671 million (12% of the global adult population) had obesity [2]. Obesity has been associated with numerous adverse health outcomes, including increased incidence of type II diabetes, cancers, and cardiovascular disease [3], and has both direct (medical) and indirect (nonmedical) costs that pose a significant global economic burden [4]. Developing effective therapeutic strategies to treat obesity is thus a global health priority.

The endocannabinoid system (ECS) was recognized as a potential target for obesity treatment in the early 2000’s [5]. The ECS is an evolutionarily conserved lipid signalling system that has widespread involvement in nearly all physiological processes, including energy homeostasis, synaptic plasticity, and feeding behaviours [6]. The major components of the ECS include: two canonical G-protein coupled receptors, the cannabinoid type-1 (CB1) and type-2 (CB2) receptors; the endogenous ligands (endocannabinoids), which are phospholipid derivatives containing a poly-unsaturated fatty acid moiety and a polar head group, either ethanolamine in the case of arachidonoylethanolamide (anandamide; AEA) or glycerol in the case of 2-arachidonoylglycerol (2-AG); and the enzymes responsible for the synthesis and degradation of the endocannabinoids [7]. One major synthetic pathway for 2-AG involves the enzyme diacylglycerol lipase (DAGL), while AEA synthesis is more varied [6]. DAGL has two isoforms, DAGLα and DAGLβ, which are encoded by two different human genes, *DAGLA* and *DAGLB* [8]. In terms of degradation, fatty acid amide hydrolase (FAAH), encoded by the human *FAAH* gene, is one of the most studied ECS enzymes, as it is responsible for the majority of AEA metabolism and is involved to a lesser extent in 2-AG metabolism [7]. Similarly, monoacylglycerol lipase (MAGL), encoded by the human *MGLL* gene, is responsible for the majority of 2-AG metabolism [7]. See Figure 1 for an overview of the ECS.

The ECS has a multi-faceted role in the control of food intake and body weight, and it acts through both peripheral and central mechanisms. In the periphery, activation of CB1 receptors (e.g., by binding of AEA or 2-AG) promotes fat storage in adipocytes and increases lipogenesis, increases glucose uptake, and reduces satiety signals [9]. Centrally, activation of the ECS interferes with the control of hunger and satiety in multiple brain regions, including the hypothalamus and brain stem [9]. This likely involves a complex interplay between endocannabinoids and other neurotransmitters or hormones involved in hunger and satiety such as cholecystokinin (CCK), glucagon-like peptide 1 (GLP-1), and neuropeptide Y (NPY) [9]. Since the primary physiological function of the ECS is thought to shift energy balance towards energy storage, overactivation of the ECS likely contributes to obesity [5]. Numerous studies have found a positive association between circulating levels of endocannabinoids (AEA and/or 2-AG) and obesity in humans [10,11,12]. The elevated levels of endocannabinoids in human obesity are related to diet [13] and have been found to decrease after bariatric surgery [14].

Given that activation of the ECS increases food intake and energy storage, and that excessive activation of the ECS contributes to the pathogenesis of obesity, antagonism of the ECS has been explored as a potential therapeutic strategy [5,15]. In the early to mid-2000’s, the multi-center Rimonabant in Obesity (RIO) trials found significant evidence that rimonabant, a CB1 receptor antagonist/inverse agonist, reduced body weight in adults with obesity, and improved other secondary measures such as dyslipidemia and cardiometabolic risk factors [16,17]. Unfortunately, the promise of rimonabant as an obesity treatment was short-lived, as it was withdrawn from the market due to mounting evidence of serious psychiatric adverse effects, including increased risk of suicidal ideation and depression [18,19,20]. The promise of rimonabant to treat obesity led to efforts to develop and test numerous other CB1 receptor ligands that might have potential to modulate body weight and treat obesity. Interested readers are directed to our previous review of CB1 receptor ligands and their potential to treat obesity [15]. In brief, some potential strategies to target the CB1 receptor without producing serious psychiatric adverse effects include peripherally-restricted CB1 inverse agonists, CB1 receptor partial agonists, and CB1 receptor neutral antagonists.

Rather than directly targeting the CB1 receptor, another potential strategy to modulate ECS activity is to target the enzymes responsible for synthesis and/or degradation of the endocannabinoids. Two approaches can be undertaken: (1) targeting endocannabinoid degradation, i.e., by inhibiting FAAH or MAGL, and (2) targeting endocannabinoid biosynthesis, i.e., by inhibiting DAGL or another synthetic enzyme [8]. Given that obesity is associated with enhanced endocannabinoid tone, inhibiting endocannabinoid biosynthesis (e.g., with a DAGL inhibitor) is the most biologically plausible approach. However, the activity of ECS enzymes is complex, so it is hard to predict the effect of inhibiting ECS enzymes. For example, in some tissues FAAH inhibition may target 2-AG in addition to AEA, and it can both increase and decrease 2-AG concentrations [21]. There are also other substrates of FAAH that have a role in metabolism and satiety such as oleoylethanolamine (OEA) and palmitoylethanoamine (PEA). OEA is notable in particular as it has been shown to reduce food intake and suppress appetite, which is opposite to the effects of the endocannabinoids [22]. As FAAH metabolizes both pro-appetitive and anti-appetitive lipids, inhibition of FAAH would be expected to have mixed effects on appetite and body weight. It should be also noted that AEA and 2-AG have complex regulations; for example, AEA has been shown to inhibit metabolism and actions of 2-AG in the striatum [23].

To date, previous reviews have highlighted the genetic association between variation in the *FAAH* gene and obesity-related outcomes in humans [9,24]. However, no previous reviews have focused specifically on evidence for a potential role of ECS enzymes in obesity treatment. Thus, the goal of the present review was to systematically review the published empirical literature and qualitatively synthesize existing evidence that either pharmacological or genetic manipulation/variation of FAAH, MAGL, and DAGL might impact obesity-related outcomes. Since the published literature we identified for the present review was so heterogeneous in study design and outcomes, we decided to present this as a narrative review.

## 2. Search Strategy

A search was conducted on 13 July 2021 in the PubMed database. The following terms were used in a Boolean search format: (FAAH OR “Fatty acid amide hydrolase” OR MAGL OR MGL OR “Monoacylglycerol lipase” OR DAGL OR DGL OR “Diacylglycerol lipase”) AND (obesity). A single search was conducted that included both preclinical and clinical studies.

Inclusion of an article was determined on the basis of its design, intervention(s), and endpoint(s). In terms of design, only studies involving humans or animal models were included. With regards to intervention, the study needed to feature (1) administration of a FAAH, MAGL, or DAGL inhibitor AND/OR (2) investigation of genetic variation in *FAAH*, *MGLL*, or *DAGLA* genes. With either of these interventions, a connection to obesity-related endpoints was also necessary. Specifically, relevant endpoints included measures of (1) body weight or body mass index (BMI), (2) body fat composition or distribution, (3) food intake or feeding behaviour, (4) glycemic markers, and/or (5) lipid profiles. These endpoints were selected from a review on metabolic endpoints in animal models of obesity [25]. Studies were included only when the title and/or abstract made an explicit mention of obesity, body weight, or related endpoints.

A hierarchy of exclusion criteria was also established. Studies were first excluded if they were written in the wrong language (i.e., not English). Next, studies were excluded on the basis of wrong study design. Common studies excluded under this criterion were review/non-empirical articles, in vitro studies, and genetic studies that did not consider variants or polymorphisms in the relevant genes. Studies were then excluded if they used the wrong intervention. Namely, this applied to pharmacological studies that did not administer a FAAH, MAGL, or DAGL inhibitor. Finally, studies were excluded if they did not measure an obesity-related outcome.

Figure 2 provides a Preferred Reporting Items for Systematic Reviews and Meta-Analyses (PRISMA) flowchart of the screening process for this review, conducted through Covidence. The initial search on July 13 produced 1235 studies from the PubMed database. During title and abstract screening, two reviewers were assigned to each article, and 1160 studies were removed. Seventy-five studies were left for full-text eligibility screening, and in this phase, 30 studies were excluded: 1 for wrong language, 18 for wrong study design, 7 for wrong intervention, and 4 for wrong outcomes. Ultimately, 45 studies were included for data extraction. During full-text eligibility screening, each article was also reviewed by two reviewers.

## 3. Current Evidence for a Role of Modulating FAAH, MAGL, and DAGL Activity in Obesity-Related Outcomes

### 3.1. Animal Studies Involving Pharmacological Manipulation

We identified 5 animal studies that administered a pharmacological inhibitor of FAAH or DAGL and measured an obesity-related outcome (see Table 1).

Balsevich et al. (2018) investigated the role of FAAH in hypophagic responses to leptin [26]. Following a period of food deprivation, leptin administration induced expected reductions in body weight gain and food intake in saline-treated mice. Animals pre-treated with the FAAH inhibitor URB597 did not show these satiety responses following leptin treatment, suggesting that FAAH inhibition suppresses the effects of leptin on body weight and food intake. In another study, pharmacological inhibition of FAAH with PF-3845 had no effect on energy storage in rats [27]. Following exposure to a high-fat diet, PF-3845 and saline-treated animals showed similar decreases in body weight and food intake.

Stearoyl-CoA desaturase-1 (SCD1) is an enzyme involved in the synthesis of endogenous monounsaturated fatty acids (MUFAs). Due to its role in obesity, Liu et al. (2013) investigated functional interactions between SCD1 and ECS activity and found that MUFAs act as endogenous inhibitors of FAAH [28]. Under a high-fat diet, *SCD1* knockout mice remained insulin-sensitive and glucose-tolerant compared to wildtype mice. When SCD1-deficient mice were treated with the FAAH inhibitor URB597, they became insulin resistant and showed increased sensitivity to glucose. Endogenous FAAH inhibition resulting from the SCD1-mediated production of MUFAs may thus contribute to the hyperinsulinemic phenotype in response to high-fat diets.

Two studies also investigated the use of DAGL inhibitors with obesity-related outcomes. Bisogno et al. (2013) found that the DAGLα inhibitor O-7460 significantly and dose-dependently reduced food intake in male mice under a high-fat diet [29]. The highest dose of O-7460 (12 mg/kg) was also associated with a small yet significant decrease in body weight. Palma-Chavez et al. (2019) used a shRNA DAGLα-inhibiting adenovirus to target tancytes of the hypothalamus and observed expression patterns of orexigenic (NPY) and anorexigenic (POMC) neuropeptides [30]. In fasting conditions, DAGLα inhibition was associated with reduced orexigenic and increased anorexigenic neuropeptides. Following glucose administration, DAGLα inhibition increased orexigenic and decreased anorexigenic neuropeptide expression. Control rats showed opposite responses in both conditions. Taken together, these findings indicate that DAGLα may regulate feeding behavior through the modulation of tancytic neuropeptides. DAGLα inhibition may reduce feeding through resulting decreases in 2-AG.

### 3.2. Animal Studies Involving Genetic Manipulation

We identified 11 preclinical studies involving the genetic manipulation of *Faah*, *Mgll*, or *Dagla* that reported obesity-related outcomes (see Table 2). All studies used C57BL/6 mouse models.

The genetic deletion of *Faah* consistently produced obesity-related phenotypes across studies, as expected from resulting increases in AEA signaling. Greater body weight, fat mass, and triglyceride levels were observed in *Faah* knockout mice compared to wildtype under both standard and high-fat diets [31,32]. *Faah*-deficient mice also showed hepatic insulin resistance, which was associated with elevated liver triglyceride and diacylglycerol content [33]. The obesity phenotype found by Vaitheesvaran et al. (2012) in *Faah*-deficient mice was accompanied with greater total food intake under a regular diet [32]. In contrast, Touriño et al. (2010) found no effect of *Faah* deletion on food intake under either a standard or a high-fat diet [31]. However, *Faah* knockout mice showed greater reinforcement and motivational responses to food.

The *FAAH* C385A polymorphism is a common loss-of-function mutation in humans. The A-allele results in lower FAAH expression and has been associated with heightened obesity risk. Balsevich et al. (2018) generated a knock-in model of this variant and found impaired leptin sensitivity in A385A mice [26]. The expected reductions in food intake and body weight following leptin treatment seen in wildtype were absent in transgenic mice, suggesting a role of FAAH in leptin satiety responses.

In addition to hydrolyzing N-acylethanolamines (NAEs) such as AEA, FAAH terminates the signaling activity of N-acyl taurines (NATs). Grevengoed et al. (2019) engineered a *Faah* knock-in model that selectively impaired NAT catabolism without affecting NAE degradation [34]. The S268D substitution did not replicate the obesity phenotypes found in the other studies. Instead, transgenic mice showed an improved metabolic profile compared to wildtypes. These findings confirm that obesity phenotypes found in *Faah*-deficient models are the result of endocannabinoid accumulation rather than increased NATs. Elevated NAT levels may instead have beneficial effects on energy storage and metabolism, which remain to be investigated.

Five studies investigated the role of *Mgll* in energy storage and obesity-related measures. Overexpression of *Mgll* in forebrain neurons lowered endocannabinoid levels, causing expected reductions in weight gain and adiposity [35]. Interestingly, intestinal overexpression of *Mgll* produced the opposite phenotype, characterized by increased body fat mass and weight gain [36]. These contradicting findings may reflect tissue-specific functions of MAGL. Similarly, under an obesogenic diet, the genetic deletion of *Mgll* was associated with paradoxical improvements in metabolic function. Compared to wildtypes, *Mgll*-deficient mice experienced less weight gain, had lower plasma triglyceride levels, and lower hepatic triglyceride content [37,38]. Insulin sensitivity and glucose tolerance were also improved [38]. Under a low-fat diet, knockout mice had lower fat mass and body weight despite showing no significant differences in food intake [39]. Reduced triglyceride and cholesterol levels were also found in male *Mgll* knockout mice, under both low and high-fat diets. When males and females were considered together, this effect was only found in response to a high-fat diet. The potential sex-specific effects of *Mgll* function on energy storage were not investigated by the other studies, as they included only male mice.

Our search identified only one genetic manipulation study on *Dagla* and obesity-related measures. Powell et al. (2015) found that *Dagla* deletion improved metabolic function [40]. Compared to wildtype, knockout mice had reduced body weight, fat mass, total triglycerides and cholesterol. Food intake and fasting insulin levels were also reduced in *Dagla*-deficient mice.

### 3.3. Human Studies Involving Genetic Association

We identified 30 studies that included association between a *FAAH* or *MGLL* gene variant and any obesity-related outcome in a human sample (see Table 3). No studies were identified involving *DAGLA* or *DAGLB* gene variants. All 30 studies included at least one *FAAH* variant, with only 3 studies additionally examining variation in *MGLL*.

The majority of studies (26 out of 30) included one specific *FAAH* SNP, a missense mutation (C385A) associated with a change in amino acid sequence at position 129 from proline to threonine (Pro129Thr). This SNP was originally identified in 2002 and was found to be strongly associated with problematic substance use [41]. The Pro129Thr variant was found to have normal catalytic activity but an enhanced sensitivity to proteolytic degradation [41]. Subsequent functional characterization of the variant found that the mutant *FAAH* enzyme had less than half of the expression and activity of the wildtype enzyme [42].

Of the 26 studies included in the present review, 19 found a significant association between the C385A SNP and at least one obesity-related outcome [43,44,45,46,47,48,49,50,51,52,53,54,55,56,57,58,59,60,61]. Following up on their initial discovery of the C385A SNP, Sipe and colleagues were the first to identify a significant association between the A allele and increased likelihood of being overweight or obese in a sample of 1688 White adults and 614 Black adults (but not in 365 Asian adults) [57]. In support of these findings, Monteleone et al. (2008) found that the A allele was significantly more common in women who were overweight or obese compared to healthy weight controls (*n* = 299) [55]. Yagin et al. (2019) similarly found the A allele to be more common in women who were overweight or obese (*n* = 180) compared to health weight controls (*n* = 86) [60]. Thethi et al. (2020) recently reported another significant association between the A allele and obesity, though the association was no longer significant when controlling for age, race, sex, waist-to-hip ratio, and cholesterol [58]. Further support for an association between the C385A variant and increased risk of obesity comes from studies including continuous outcomes. For example, the A allele was associated with higher BMI, fat mass, and waist circumference in a sample of 70 participants with Type-II diabetes mellitus who were obese [50]. Yagin et al. (2019) also found a significant association between the A allele and higher BMI, waist circumference, neck circumference, waist-to-height ratio, and body fat mass [60], while Zhang et al. (2009) found an association between the A allele and higher BMI and triglyceride levels in a sample of 1644 participants from 261 pedigrees [61]. Further, Vazquez-Roque et al. (2011) found that A-allele carriers had a significantly greater maximum tolerated volume after a nutrient drink test, indicating lower satiety [59].

In contrast, multiple studies have failed to replicate the association between the A allele and obesity. Seven studies found no significant association between the SNP and any obesity-related outcomes [62,63,64,65,66,67,68]. A few studies actually found the reverse association—either a significant association between the wildtype (C/C) genotype and obesity [53,56] or an association of the wildtype (C/C) genotype with worse cardiometabolic outcomes, e.g., higher triglycerides and metabolic biomarkers [46]. Studies conducted by De Luis and colleagues failed to find any significant association between the C385A SNP and a wide range of obesity-related outcomes including anthropometric outcomes (BMI, body weight, waist circumference, etc.) and circulating lipids [44,45,46,47,48,49,51,52].

Interestingly, a number of studies have suggested that while the C385A SNP may not directly impact obesity-related outcomes at baseline, there may be an impact of the variant on changes in outcomes following either surgical or lifestyle interventions for obesity. Aberle et al. (2007) found that A-allele carriers had greater decreases in triglyceride and cholesterol levels after a 6-week low-fat diet intervention (*n* = 451 obese and dyslipidemic participants) [43]. Similarly, De Luis et al. (2011) found that A-allele carriers had greater improvement in a range of metabolic outcomes after 3 months of a hypocaloric dietary intervention (*n* = 122) [47], while De Luis et al. (2010) found greater weight loss in A-allele carriers at 9 and 12 months after biliopancreatic diversion surgery (*n* = 67) [51]. In contrast, three other studies by De Luis and colleagues found that the A-allele was associated with worse outcomes following either a low-fat or low-carbohydrate diet intervention for 3 months (*n* = 248) [52], and also following a 3-month enriched monounsaturated fat hypocaloric diet (*n* = 95) [44] and a 3-month enriched polyunsaturated fat hypocaloric diet (*n* = 99) [49]. Finally, Knoll et al. (2012) failed to show any association of the C385A variant and obesity-related outcomes after a 1-year diet and exercise intervention in a group of 453 overweight and obese children and adolescents [63].

In addition to examining associations between the *FAAH* C385A variant and obesity-related outcomes, four studies examined associations with other *FAAH* SNPs [53,56,61,64]. In a large sample of 5109 French Caucasian participants, Durand et al. (2008) failed to find an association between 10 *FAAH* SNPs and childhood obesity or Type-II diabetes mellitus, but did find a nominal association between 5 *FAAH* SNPs (rs6429600, rs324419, rs324418, rs2295633, and rs7520850) and Class III adult obesity [53]. Lieb et al. (2009) failed to find an association between 9 *FAAH* SNPs and obesity-related outcomes in a sample of 2415 participants from a longitudinal cohort [64]. Muller et al. (2010) found some additional evidence of association between *FAAH* gene variants and obesity—the G allele of rs2295632 was associated with early-onset obesity in 521 trios (children/adolescents and both biological parents), though this finding did not replicate in a second cohort of trios, and there was no association with obesity in adults (*n* = 8491 participants from a population-based study sample) [56]. Finally, Zhang et al. (2009) did not find an association between four *FAAH* SNPs (rs324418, rs1984490, rs2145408 and rs4141964) and any obesity-related traits [61].

Three studies used bioinformatic approaches to identify novel genetic variants in *FAAH* and *MGLL* genes associated with obesity or obesity-related traits [69,70,71]. Bhatia et al. (2010) used data from a large clinical trial cohort (*n* = 289) to identify rare variants in *FAAH* and *MGLL* associated with obesity, and found one significantly associated region of approximately 5 Kbp in the upstream regulatory region of each gene [69]. Harismendy et al. (2010) similarly used data from the same clinical trial cohort to identify one interval in the *FAAH* promoter and three intervals in *MGLL* (in the promoter, intron 2, and intron 3) that were all significantly associated with obesity [70]. Kuk et al. (2013) identified three potentially causal rare variants in *MGLL* associated with obesity, while also finding an interaction between two rare variants in *FAAH* that may increase the risk of obesity [71].

## 4. Discussion

The endocannabinoid system plays an important role in the regulation of hunger, satiety, and body weight. Obesity is associated with chronic over-activation of the ECS, with increased levels of circulating endocannabinoids. Thus, modulating the activity of ECS enzymes, thereby altering endocannabinoid signaling, is a biologically plausible strategy for treating obesity. Our review focused on obesity-related outcomes associated with manipulation of or variation in three key enzymes: FAAH, an enzyme responsible for degradation of many bioactive lipids, especially AEA; MAGL, another enzyme responsible for degradation, especially 2-AG; and DAGL, an enzyme involved in the synthesis of 2-AG. We identified 15 studies that involved pharmacological or genetic manipulation of at least one of these enzymes and assessment of at least one obesity-related outcome in an animal model, and 30 studies that examined a genetic association between variants in the genes encoding FAAH and MAGL and obesity in humans. Overall, we found strong evidence that modulating ECS enzyme activity can impact obesity-related outcomes, but the nature of the association is complex and likely dependent on numerous factors, such as species, age, sex, diet, and physical activity.

The preclinical studies reviewed found that either pharmacological inhibition or genetic deletion of FAAH produced obesogenic phenotypes, generally increased body weight, and/or worse metabolic outcomes. For example, *Faah* knockout mice had greater body weight, fat mass, and triglyceride levels compared to wildtype mice, under both a standard and high-fat diet [31,32]. It is important to note that there are many substrates of FAAH other than AEA (many of which do not act at CB receptors) and that the functions of FAAH substrates are incredibly diverse. Genetic deletion or pharmacological inhibition of FAAH produces a myriad of behavioural effects in animal models, including modulation of anxiety- and depressive-like behaviours, alterations in gastrointestinal function, decreases in pain response, decreases in inflammation, and modulation of drug-seeking behaviours and withdrawal [8]. As a result, FAAH inhibitors have been investigated in preclinical models of an incredibly wide range of pathological conditions, including pain (e.g., inflammatory, neuropathic, cancer-associated), neurological conditions (e.g., traumatic brain injury, epilepsy, movement-related disorders), gastrointestinal disorders, cardiovascular disorders, and psychiatric disorders (including substance use disorders) [21].

Of the human genetic association studies reviewed, most (26/30) focused on the rs324420 SNP of *FAAH*. Despite the initially compelling association between the A allele of this SNP and increased likelihood of being overweight and obese in a large sample of white and Black adults by Sipe and colleagues [57], which was replicated, at least in part, in three more recent studies [55,58,60], multiple studies have failed to find any association between rs324420 genotype and obesity [62,63,64,65,66,67,68]. In addition, two studies actually found a significant association between the homozygous major allele genotype (C/C) and obesity [53,56]. Of note, a handful of studies found that carriers of the A allele of rs324420 had greater improvements after either lifestyle (diet and/or exercise) intervention [43,47] or surgical intervention [51]. However, other studies failed to find greater improvement in A allele carriers [63] or found greater improvement associated with the major C/C genotype [44,49].

The effect of genetic manipulation of *Mgll* appears to be complex, and likely tissue-dependent. Preclinical studies found that overexpression of *Mgll* in forebrain neurons caused a reduction in weight gain and adiposity [35], yet intestinal overexpression of *Mgll* actually led to increased weight gain and fat mass [36]. The tissue-specific effects of *Mgll* deletion highlight the complexity of ECS control of energy intake and body weight. For example, MAGL is involved in the hydrolysis of triacylglycerols in adipose tissue [73]. Thus, future work should consider tissue-specific manipulation of MAGL on obesity-related outcomes, which may have important implications for therapeutic trials in humans.

Studies examining manipulation of DAGL were scarce. One study found that the DAGLα inhibitor O-7460 significantly decreased body weight in male mice [29], while another found that *Dagla* knockout mice had reduced body weight, fat mass, and total triglycerides compared to wildtype mice [40]. No human studies were identified that examined associations between *DAGLA* gene variants and obesity. This very preliminary evidence suggests that reducing biosynthesis of 2-AG could be one strategy to recapitulate the therapeutic effects of CB1 receptor antagonism, ideally without the serious psychiatric adverse effects as seen with rimonabant. Similarly, it could be worth exploring potential inhibitors of AEA biosynthesis. To date, there is no single enzyme purported to be the single “AEA synthase”, though N-acetylphosphatidylethanolamine-hydrolysing phospholipase D (NAPE-PLD) has been proposed as a candidate [8]. Interestingly, deletion of the gene encoding NAPE-PLD in mouse adipocytes induced an obese state characterized by glucose intolerance, adipose tissue inflammation, and altered lipid metabolism [74]. However, NAPE-PLD is involved in the biosynthesis of many other similar lipids, and NAPE-PLD genetic knockout does not consistently lead to a reduction in AEA [8].

While our review considered only candidate gene association studies in humans involving *FAAH* or *MGLL*, a brief note on genome-wide association studies (GWAS) of obesity is worthwhile. Small-scale candidate gene associations studies, such as some of the studies reviewed here, have been useful to identify specific monogenic forms of obesity, which are inherited in a Mendelian pattern and are typically rare, severe, with early onset [75]. Such approaches can also be used to identify common variants associated with polygenic (common) obesity. For example, candidate gene association approaches using larger samples were used to identify significant associations of variants in the CB1 receptor gene (*CNR1*) with obesity outcomes, which has been a reproducible association [76]. However, GWAS approaches using very large population samples are necessary to confirm the association of common variants contributing small effects to obesity [75]. Loos and Yeo (2021) extracted GWAS data from GWASCatalog, spanning from the first obesity GWAS in 2007 to 25 January 2021. A review of these data found no SNPs in *FAAH*, *MGLL*, or *DAGLA*/*DAGLB* that showed genome-wide significant association (*p* < 5 × 10^−8^) with obesity or related outcomes (also of note, no SNPs in *CNR1* either) [75]. This suggests that variation in ECS genes is unlikely to make a major contribution to polygenic obesity, which is in line with the heterogeneity in results of candidate gene association studies (*FAAH* and obesity outcomes) we reviewed here.

Following up on the promising findings of preclinical animal studies of FAAH inhibition in multiple disease models, a number of human trials have been conducted or are ongoing to evaluate the potential of FAAH inhibitors across a range of indications [21]. For example, recently completed trials have found a positive signal for the FAAH inhibitor JNJ-42165279 in reducing anxiety symptoms in patients with social anxiety disorder [77] and the FAAH inhibitor PF-04457845 in reducing cannabis withdrawal in men with cannabis dependence [78]. A search of ClinicalTrials.gov identified a number of registered ongoing trials of FAAH inhibitors for substance use disorders, Tourette Syndrome, and pain. There have also been MAGL and dual FAAH/MAGL inhibitors developed to be evaluated in controlled trials. While our review has not provided clear evidence that FAAH or MAGL inhibitors would be useful for treating obesity, it will be worth noting whether there are any weight- or obesity-related effects observed in FAAH and/or MAGL inhibitor trials.

One final important consideration is the role of sex in modulating cannabinoid action. A wealth of preclinical data have established that sex significantly impacts the role of cannabinoids in multiple physiological processes, including pain, reward and addiction, and hunger and energy homeostasis [79,80]. For example, it has been demonstrated in rodent models that estradiol replacement in ovariectomized females tends to reduce energy intake in response to the CB1 receptor agonist WIN 55,212-2, while testosterone replacement in orchidectomized males tends to increase energy intake, which can be blocked by the CB1 receptor antagonist AM251 [80]. Unfortunately, the majority of the preclinical studies we reviewed conducted experiments in male animals only. Given that estrogens and testosterone may have divergent effects on energy intake (at least in rodents), future studies must consider sex as a biological variable in order to better understand how manipulation of the ECS may impact obesity differently in male and female bodies.

Taken together, the reviewed body of literature provides a mixed picture of whether FAAH, MAGL, or DAGL are likely to be druggable targets for obesity treatment. Studies in rodent models showed that reduction in FAAH activity leads to increased body weight and metabolic function, and while the human genetic association studies found consistent evidence of association between *FAAH* variants and obesity, the direction of effect was inconsistent. Thus, based on the evidence reviewed, FAAH inhibition is not likely a promising approach for obesity treatment. It will be prudent to monitor changes in body weight in ongoing trials of FAAH inhibitors for other indications. The role of MAGL modulation in obesity-related phenotypes appeared to be tissue dependent, with different effects depending on whether central or peripheral MAGL was targeted. Further research is needed to clarify this association. Based on a handful of animal studies, inhibition of DAGLα may be a promising strategy to treat obesity. This approach is in line with the higher levels of circulating endocannabinoids found in animals and humans with obesity. However, given the complex relationship between ECS activity and obesity, more evidence is needed to characterize the role of DAGL modulation in obesity. Human studies evaluating associations between *DAGLA* (and possibly also *DAGLB*) variants and obesity may provide some additional insight into whether modulating 2-AG synthesis impacts obesity-related outcomes.

## Figures and Tables

**Figure 1 pharmaceuticals-14-01316-f001:**
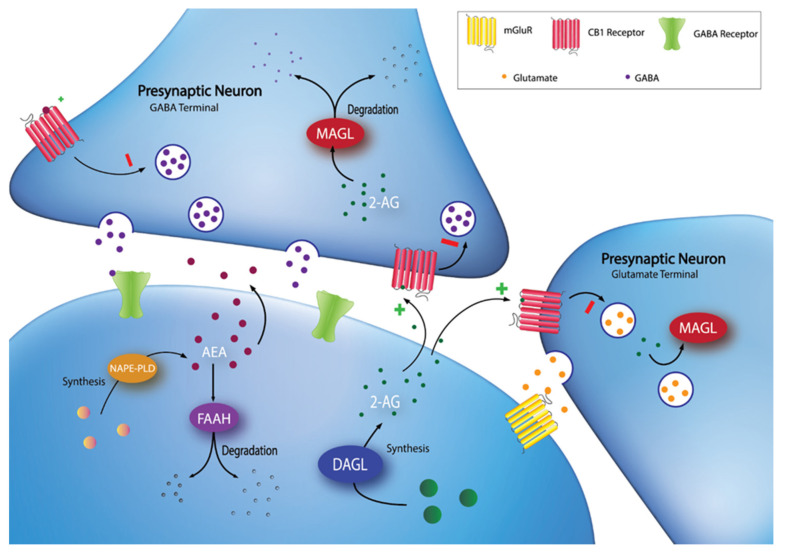
Synaptic localization of endocannabinoid synthesis and degradation. 2-AG is synthesized on-demand in the postsynaptic neuron by DAGL. Following synthesis, 2-AG diffuses into the synaptic cleft and activates CB1 receptors at GABA and glutamate terminals. 2-AG signaling can be terminated by MAGL degradation in the presynaptic terminal. AEA can be synthesized through multiple pathways (e.g., through a pathway involving NAPE-PLD), then diffuses into the synapse to activate CB1 receptors. Extracellular AEA undergoes reuptake into the postsynaptic cell and is hydrolyzed by FAAH.

**Figure 2 pharmaceuticals-14-01316-f002:**
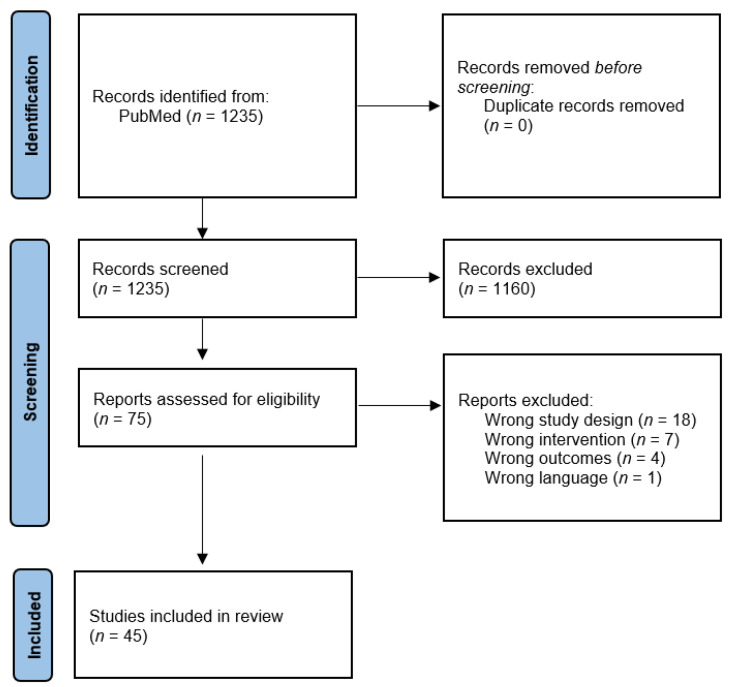
Flow diagram depicting the process of article identification, screening, and inclusion according to Preferred Reporting Items for Systematic Reviews and Meta-Analyses (PRISMA) guidelines. Both preclinical and clinical studies were included in the same article screening process.

**Table 1 pharmaceuticals-14-01316-t001:** Studies in animal models evaluating the impact of pharmacological manipulation of FAAH or DAGL enzymes on obesity-related outcomes.

Reference	Species/Strain and Sex	Drug, Dose, and Route of Administration	Key Findings
**FAAH Inhibitors**			
**Balsevich et al. (2018) [26]**	C57BL/6 mice, male	0.3 mg/kg of URB597 i.p.	After a 12-h overnight fast followed by treatment with URB597, leptin administration was unable to significantly reduce body weight gain or food intake
**Cifani et al. (2020) [27]**	Sprague-Dawley rats, male	10 mg/kg of PF-3845 i.p.	No significant impact on body weight
**Liu et al.** **(2013) [28]**	C57BL/6 and SV/129 mice, male	5 mg/kg of URB597 i.p	*SCD1*^−/−^ mice fed a HFD demonstrated glucose intolerance and insulin resistance after treatment with URB597 *SCD1*^−/−^ mice fed a HFD had significantly higher levels of plasma insulin after URB597 administration
**DAGL Inhibitors**			
**Bisogno et al. (2013) [29]**	C57BL/6 mice, male	O-7460 i.p.; 0 mg/kg, 6 mg/kg, and 12 mg/kg	O-7460 dose-dependently decreased intake of a HFD over 14 h O-7460 (12 mg/kg) significantly decreased body weight at 14 h
**Palma-Chavez et al. (2019) [30]**	Sprague-Dawley rats, female	30 μL (at 5 × 10^7^ IFU/mL) of shRNA DAGLα-inhibiting adenovirus; administered into third ventricle	In fasting conditions, inhibition of DAGLα reduced NPY and increased POMC expression; in response to glucose, DAGLα inhibition increased NPY expression but decreased POMC (opposite in control rats)

DAGL, diacylglycerol lipase; FAAH, fatty acid amide hydrolase; HLD, high-fat diet; i.p., intraperitoneal; NPY, neuropeptide Y; POMC, proopiomelanocortin.

**Table 2 pharmaceuticals-14-01316-t002:** Studies in animal models evaluating the impact of genetic manipulation of *Faah* or *Mgll* on obesity-related outcomes.

Reference	Species/Strain and Sex	Gene(s) of Interest and Genetic Manipulation *	Key Findings
**Balsevich et al. (2018) [26]**	C57BL/6 mice, male	Mouse line with *Faah* C385A knock-in	After a 16-h overnight fast, C385A knock-in mice did not demonstrate an expected decrease in cumulative food intake or body weight after leptin administration
**Brown et al. (2012) [33]**	129SvJ-C57BL/6 mice, both male and female	*Faah* ^−/−^ mice	*Faah*^−/−^ mice had significantly high levels of liver triglycerides and liver diacylglycerols compared to WT mice Deletion of *Faah* resulted in ectopic lipid accumulation at the liver
**Chon et al. (2012) [36]**	C57/BL6J-SJL mice, male	Small intestine *Mgll* was overexpressed in mice (iMGL mice)	After three weeks of a HFD, iMGL mice (vs. WT) had significantly greater weight gain, increased percent body fat, increased adipose tissue mass, greater ectopic fat depositions in the liver and small intestine, and increased foot intake with reduced energy expenditure
**Douglass et al. (2015) [39]**	129/SvEv-C57BL/6 mice, both male and female	*Mgll*^−/−^ mice	At baseline, *Mgll* ^−/−^ and WT mice did not differ in body weight Throughout 12 weeks of feeding, *Mgll* ^−/−^ mice had lower body weight compared to WT mice, in both males and females, and under both HFD and low-fat diets LFD At baseline and after 12 weeks of low-fat dieting, *Mgll* ^−/−^ mice had a significantly lower fat mass than WT mice Throughout the 12-week feeding period, there were no significant differences in cumulative or average daily food intake; after the feeding period, LFD-fed male *Mgll* ^−/−^ mice had a significantly higher cumulative food intake Male *Mgll* ^−/−^ mice had significantly lower triglyceride and cholesterol levels under both diets; when considering both sexes, only *Mgll* ^−/−^ fed a HFD demonstrated significantly lower triglyceride and cholesterol levels
**Grevengoed et al. (2019) [34]**	C57BL/6N mice, male	Single amino-acid substitution in *Faah* (S268D) that selectively disrupts *N-*acyl taurine (NAT) but not *N-*acylethanolamine (NAE) hydrolytic activity	*Faah*-S268D mice had improved insulin sensitivity, increased glucagon secretion after insulin, and lower food intake in response to leptin treatment compared to WT mice
**Jung et al. (2012) [35]**	C57BL/6J mice, male	*Mgll* gene; generation of mice with forebrain neurons overexpressing *Mgll* (using CaMKIIα promoter)	*Mgll*-overexpressing mice showed reduced weight gain, decreased adiposity, and increased lean mass compared to WT mice; this was accompanied by lower plasma triglyceride levels, decreased serum glucose levels, and increased glucose uptake despite increased frequency of feeding and greater food intake *Mgll*-overexpressing mice were resistant to diet-induced obesity compared to WT
**Powell et al. (2015) [40]**	129S5/SvEvBrd x C57BL/6-Tyr^c−Brd^ mice, both male and female	*Dagla* and *Daglb* genes; generation of *Dagla* KO, *Daglb* KO, and *Dagla/Daglb* double KO mice	*Dagla* KO mice had significantly lower body weight and body fat mass compared to WT mice (under both chow-fed and HFD) *Dagla* KO mice had significantly lower total food intake compared to WT mice *Dagla* KO mice had significantly lower fasting insulin, total triglyceride, and total cholesterol compared to WT mice
**Tardelli et al. (2019) [37]**	C57BL/6 mice, male	*Mgll* ^−/−^ mice	Under either chow or Western diet, *Mgll* ^−/−^ mice experienced significantly less weight gain compared to WT mice, but *Mgll* ^−/−^ mice had a significantly greater gonadal white adipose tissue to body weight ratio; no significant differences in food intake *Mgll* ^−/−^ fed under a Western diet had significantly lower levels of plasma and hepatic triglycerides and plasma insulin compared to WT mice; no significant differences in plasma cholesterol
**Touriño et al. (2010) [31]**	129SvJ-C57BL/6 mice, male	*Faah* ^−/−^ mice	*Faah*^−/−^ mice had significantly greater body weight (standard diet and high-fat diet) compared to WT mice; this difference increased over time in *Faah*^−/−^mice fed with the HFD No significant effect on total food intake; however, *Faah*^−/−^ mice demonstrated greater reinforcement and motivation effects from food When fed a HFD, *Faah*^−/−^ mice had significantly higher fat mass, triglyceride levels, glucose, and insulin levels compared to WT mice
**Vaitheesvaran et al. (2012) [32]**	C57BL/6 mice, male	*Faah* ^−/−^ mice	*Faah* ^−/−^ mice had significantly higher body weight, food intake (regular chow diet), and fat mass compared to WT mice *Faah* ^−/−^ mice had significantly higher levels of plasma insulin and plasma triglycerides compared to WT mice
**Yoshida et al. (2019) [38]**	C57BL/6 mice, male	*Mgll* ^−/−^ mice	*Mgll* ^−/−^ and WT mice had similar body weight gain and food intake when fed a normal chow diet; however, when fed a HFD, *Mgll* ^−/−^ mice gained less weight compared to WT mice *Mgll* ^−/−^ mice had significantly better glucose tolerance and insulin sensitivity compared to WT mice *Mgll* ^−/−^ mice had significantly lighter liver weights compared to WT mice, indicative of less ectopic fat accumulation in the liver Following oral gavage with olive oil, *Mgll* ^−/−^ mice had significantly lower plasma triglyceride levels compared to WT mice

* All gene knock-outs are global knock-outs unless otherwise indicated. Dagla, diacylglycerol lipase α; Daglb, diacylglycerol lipase β; Faah, fatty acid amide hydrolase; HLD, high-fat diet; KO, knock-out; LFD, low-fat diet; Mgll, monoacylglycerol lipase; WT, wildtype.

**Table 3 pharmaceuticals-14-01316-t003:** Human studies examining genetic association between *FAAH* and/or *MGLL* and obesity-related outcomes.

Reference	Sample	Gene(s) and Variant(s) and Study Design	Key Findings
**Aberle et al. (2007) [43]**	*n* = 451 obese/overweight (BMI > 25 kg/m^2^) and dyslipidemic	*FAAH*—Pro129Thr SNP (rs324420) Genetic association with obesity-related traits at baseline and after a 6-week diet intervention	No baseline association After six weeks of low-fat diet, there were significant decreases in triglyceride and total cholesterol levels in CA/AA vs. CC
**Bhatia et al. (2010) [69]**	*n* = 147 with BMI < 30 kg/m^2^ *n* = 142 with BMI > 40 kg/m^2^	A novel algorithm called RareCover used to analyze the contribution of rare genetic variants in *FAAH* and *MGLL* to the development of obesity phenotypes (based on BMI)	Two 5 Kbp regions in the upstream regulatory segments of *FAAH* and *MGLL* genes identified, rare variants that could hinder *FAAH* and *MGLL* gene expression
**De Luis et al. (2010) [46]**	*n* = 279 obese females	*FAAH*—Pro129Thr SNP (rs324420) Genetic association with obesity-related traits	No significant association with anthropometric parameters or dietary intake A-allele significantly associated with lower triglycerides, glucose, and HOMA
**De Luis et al. (2010) [48]**	*n* = 143 obese females	*FAAH*—Pro129Thr SNP (rs324420) Genetic association with obesity-related traits	No significant differences in anthropometric parameters or food intake Significantly higher levels of glucose, insulin, and HOMA in C/C (wildtype) vs. A-allele carriers
**De Luis et al. (2010) [50]**	*n* = 70 with Type-II Diabetes Mellitus and obesity (BMI > 30 kg/m^2^)	*FAAH*—Pro129Thr SNP (rs324420) Genetic association with obesity-related traits	A-allele carriers had significantly higher BMI, fat mass, waist circumference, insulin, HOMA, and lower adiponectin vs. wildtype (C/C) No significant difference in dietary intake
**De Luis et al. (2010) [51]**	*n* = 67 with BMI > 40 kg/m^2^ who had undergone biliopancreatic diversion	*FAAH*—Pro129Thr SNP (rs324420) Genetic association with obesity-related traits at baseline, and then 3, 9, and 12 months after surgery	No significant baseline differences in anthropometric or biochemical parameters At 9 and 12 months, weight loss was greater in A-allele carriers vs. wildtype (C/C)
**De Luis et al. (2010) [52]**	*n* = 248 with BMI > 30 kg/m^2^	*FAAH*—Pro129Thr SNP (rs324420) Genetic association with obesity-related traits at baseline and after a 3-month diet intervention (low fat or low carbohydrate)	No significant difference between genotypes at baseline or 3 months in anthropometric outcomes, cardiovascular risk factors, or circulating adipocytokines In both diet intervention groups, A-allele carriers failed to show improvement in glucose, insulin, HOMA, and leptin, while wildtype (C/C) did
**De Luis et al. (2011) [47]**	*n* = 122 with BMI > 30 kg/m^2^	*FAAH*—Pro129Thr SNP (rs324420) Genetic association with obesity-related traits at baseline and after a 3-month diet intervention (hypocaloric)	At baseline, there were no genotype differences in anthropometric or dietary intake measures, but A-allele carriers had significantly lower insulin, HOMA, and C-reactive protein vs. wildtype (C/C) After 3 months, weight, waist circumference, insulin, C-reactive protein, and triglyceride levels were lower in A-allele carriers vs. wildtype (C/C), with no difference in dietary intake
**De Luis et al. (2012) [45]**	*n* = 799 with mean BMI of 36.7 kg/m^2^	*FAAH*—Pro129Thr SNP (rs324420) Genetic association with obesity-related traits and metabolic syndrome	No significant association with metabolic syndrome No significant differences in anthropometric measures or circulating adipocytokines by genotype, but insulin and HOMA were significantly higher in A-allele carriers vs. wildtype (C/C)
**De Luis et al. (2013) [44]**	*n* = 95 with BMI > 30 kg/m^2^	*FAAH*—Pro129Thr SNP (rs324420) Genetic association with obesity-related traits at baseline and after a 3-month diet intervention (enriched monounsaturated fat hypocaloric)	No significant differences at baseline between genotypes After 3 months, the wildtype group (C/C) showed greater improvements in insulin, HOMA-R, weight, fat mass, and waist circumference vs. A-allele carriers
**De Luis et al. (2013) [49]**	*n* = 99 with BMI > 30 kg/m^2^	*FAAH*—Pro129Thr SNP (rs324420) Genetic association with obesity-related traits at baseline and after a 3-month diet intervention (enriched polyunsaturated fat hypocaloric)	No significant differences at baseline between genotypes After 3 months, there were no significant genotype differences in anthropometric parameter or adipokine changes, while there was improvement in insulin and HOMA-R in wildtype (C/C) vs. A-allele carriers
**Durand et al. (2008) [53]**	*n* = 1340 healthy adult controls *n* = 635 obese children *n* = 896 adults with Class III obesity, i.e., BMI ≥ 40 kg/m^2^ *n* = 2238 adults with Type-II Diabetes Mellitus	*FAAH—*10 SNPs across the entire *FAAH* locus, including Pro129Thr SNP (rs324420); rs913168, rs17361950, rs6429600, rs324419, rs324418, rs2295633, rs11576941, rs324425, rs7520850 Case-control genetic association	No significant associations with childhood obesity or Type-II Diabetes Mellitus Nominally significant association between rs324420 (C allele was risk allele) and 5 other *FAAH* SNPs (rs6429600, rs324419, rs324418, rs2295633, and rs7520850) and Class III (adult) obesity No significant association between rs324420 SNP and metabolic traits
**Grolmusz et al. (2013) [54]**	*n* = 63 patients with polycystic ovary syndrome (mean BMI 29.6 kg/m^2^) *n* = 67 healthy controls (mean BMI 21.2 kg/m^2^)	*FAAH*—Pro129Thr SNP (rs324420) Case-control genetic association	In cases, free thyroxine was higher in A-allele carriers than wildtype (C/C) In controls, insulin was significantly lower in A-allele carriers than wildtype (C/C) No other significant associations (including BMI)
**Harismendy et al. (2010) [70]**	*n* = 147 with BMI < 30 kg/m^2^ *n* = 142 with BMI > 40 kg/m^2^	Sequence-based case-control genetic association (*FAAH* and *MGLL*)	One interval in *FAAH* promoter and 3 intervals in *MGLL* (promoter, intron 2, intron 3) were associated with obesity
**Jensen et al. (2007) [62]**	*n* = 5801 classified as normal weight = BMI 18.5–25 kg/m^2^, overweight = BMI 25–30 kg/m^2^, and obese = BMI ≥ 30 kg/m^2^	*FAAH*—Pro129Thr SNP (rs324420) Genetic association	No significant association with BMI or waist circumference groups or with any quantitative trait
**Knoll et al. (2012) [63]**	*n* = 453 overweight or obese children and adolescents (mean 10.8 years old)	*FAAH*—Pro129Thr SNP (rs324420) Genetic association with obesity-related traits at baseline and after a 1-year lifestyle intervention	No significant association with change in any outcome
**Kuk et al. (2013) [71]**	*n* = 148 with BMI > 40 kg/m^2^ *n* = 150 with BMI < 30 kg/m^2^	Sequence-based case-control genetic association (*FAAH* and *MGLL*)	Three potentially causal rare variants were identified around *MGLL* An interaction between two rare variants around *FAAH* was identified that may increase the risk of obesity
**Lieb et al. (2009) [64]**	*n* = 2415 participants from a longitudinal cohort	*FAAH—*9 SNPs across the entire *FAAH* locus, including Pro129Thr SNP (rs324420); rs12073998, rs6703669, rs3766246, rs324419, rs2295633, rs12029329, rs324425, rs7520850 Genetic association	No significant associations
**Mansouri et al. (2020) [65]**	*n* = 79 healthy participants	*FAAH*—Pro129Thr SNP (rs324420)	No significant difference in mean BMI between genotypes
**Martins et al. (2015) [66]**	*n* = 100 with BMI ≥ 18.5 kg/m^2^ and < 25 kg/m^2^ *n* = 100 with BMI ≥ 30 kg/m^2^	*FAAH*—Pro129Thr SNP (rs324420) Genetic association with obesity and insulin-resistant phenotype	No significant associations
**Monteleone et al. (2008) [55]**	*n* = 115 overweight/obese females with binge-eating disorder (BED) *n* = 74 obese females without BED *n* = 110 healthy controls	*FAAH*—Pro129Thr SNP (rs324420) Case-control genetic association	A allele was significantly more frequent in overweight/obese females compared to controls
**Muller et al. (2010) [56]**	*n* = 521 children and adolescents (mean BMI 31.86 kg/m^2^) and both biological parents *n* = 501 German children and adolescents (including one sibling) (mean BMI 32.28 kg/m^2^) and both biological parents *n* = 8491 adults (mean BMI 27.12 kg/m^2^) from a population-based study group *n* = 985 cases (mean BMI 36.04 kg/m^2^) and *n* = 588 controls (mean BMI 19.34 kg/m^2^)	*FAAH*—five SNPs, including Pro129Thr SNP (rs324420); rs324419, rs873978, rs2295632 and rs932816 Genetic association first in a sample of trios, then replicated in a second cohort of families, then in combined sample SNPs significantly associated with childhood obesity were then screened in a population-based cohort of adults, and then in a case-control sample	Association of G allele of *FAAH* rs2295632 with early onset obesity in the initial sample of trios, but this did not replicate in the subsequent sample When the two samples were combined, *FAAH* rs324420 (C allele) and rs2295632 were significantly associated with childhood obesity No significant association between *FAAH* rs324420 or rs2295632 and adult obesity or BMI in the population-based sample; no significant association with rs324420 in the case-control analysis
**Ning et al. (2017) [72]**	*n* = 227 with BMI 35.1–61.7 kg/m^2^ *n* = 219 with BMI 17.5–23.0 kg/m^2^	*FAAH*—whole-exome sequencing	The novel *FAAH* c.G944T (p.R315I) variant co-segregated with obesity in the proband’s pedigree; in vitro characterization of *FAAH*-R315I suggested that it is a loss-of-function mutation
**Papazoglou et al. (2008) [67]**	*n* = 158 with BMI > 40 kg/m^2^ *n* = 145 with BMI > 40 kg/m^2^ and metabolic syndrome *n* = 121 with BMI 18.5–25 kg/m^2^	*FAAH*—Pro129Thr SNP (rs324420) Case-control genetic association	No significant associations
**Sipe et al. (2005) [57]**	*n* = 2667 (1688 White, 614 Black, 365 Asian)	*FAAH*—Pro129Thr SNP (rs324420) Case-control genetic association	Significant association of A/A genotype with overweight/obese status in Black and White, but not Asian, participants Median BMI was higher in the A/A genotype group compared to C/A + C/C in the pooled sample
**Thethi et al. (2020) [58]**	*n* = 465 with BMI ≥ 30 kg/m^2^ *n* = 202 with BMI ≤ 27 kg/m^2^	*FAAH*—Pro129Thr SNP (rs324420) Case-control genetic association	Significant association of the A allele with obesity, but not when adjusting for age, race, sex, waist-hip ratio, and LDL No association with any other outcome measure
**Vazquez-Roque et al. (2011) [59]**	*n* = 62 adults overweight or obese (*n* = 5, mean BMI 24.0 kg/m^2^; *n* = 28, mean BMI 28.1 kg/m^2^; *n* = 29, mean BMI 34.9 kg/m^2^)	*FAAH*—Pro129Thr SNP (rs324420) Genetic association with gastric emptying (GE) of solids and liquids, gastric volume (GV), and satiation [maximum tolerated volume (MTV) after nutrient drink test]	No significant association with GE of solids or liquids, GV, or aggregate symptom score Significant association with MTV: lower MTV in the CC genotype vs. CA/AA
**Yagin et al. (2019) [60]**	*n* = 180 healthy overweight/obese women (BMI = 25–40 kg/m^2^) *n* = 86 women with BMI = 18.5–24.9 kg/m^2^	*FAAH*—Pro129Thr SNP (rs324420) Case-control genetic association	A/A and C/A genotypes were more frequent in overweight/obese women A-allele carriers had significantly higher BMI, waist circumference, neck circumference, waist-to-height ratio, and body fat mass A-allele significantly predicted risk of obesity after adjusting for age, marital status, and physical activity. (OR: 2.38; 95% CI = 1.37–3.79)
**Yagin et al. (2020) [68]**	*n* = 180 women (mean BMI 32.54 kg/m^2^)	*FAAH*—Pro129Thr SNP (rs324420) Genetic association with binge eating disorder	No significant association, though the frequency of the A allele was numerically higher in women with binge eating disorder (*p* = 0.08)
**Zhang et al. (2009) [61]**	*n* = 1644 (from 261 extended families)	*FAAH*—Pro129Thr SNP (rs324420) and four additional SNPs (rs324418, rs1984490, rs2145408 and rs4141964) Genetic association with obesity-related traits	A-allele of rs324420 was significantly associated with higher BMI and fasting triglyceride levels compared to wildtype (C/C genotype)

FAAH, fatty acid amide hydrolase; HOMA, homeostatic model assessment; LDL, low-density lipoprotein; MGLL, monoacylglycerol lipase; SNP, single-nucleotide polymorphism.

## Data Availability

Data sharing not applicable.

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
