# Peer review of "Potential of Fatty Acid Amide Hydrolase (FAAH), Monoacylglycerol Lipase (MAGL), and Diacylglycerol Lipase (DAGL) Enzymes as Targets for Obesity Treatment: A Narrative Review"

_pharmaceuticals, 2021, doi:10.3390/ph14121316_

Round 1
Reviewer 1 Report
The authors review preclinical and clinical studies that have assessed effects of inhibition or genetic manipulation in rodents, or human genetic polymorphisms of endocannabinoid metabolizing enzymes in the context of obesity.
Overall, the review is well written. The literature searching procedures followed standard procedures. It is unclear if pre-clinical and clinical papers were searched together or separately. One would expect to use different criteria of the quality and results should be presented separately. The tables reflect summarize the key results. Some of the 45 papers appear to be missing. Because of sex differences of endocannabinoid associations with obesity it would be important to dissect it. For genetic models, substructuring for general versus cell type specific genetic manipulation would be helpful (Table 2).
The abstract remains rather vague and one cannot learn a key message from the abstract which one would expect as a take-home message. The abstract does not reveal the controversial pro and anti-obesogenic properties of e.g. FAAH substrates (AEA, OEA, PEA) and the specific features of 2-AG in the CNS and peripheral tissues.
Indeed, oleoylethanolamide which is a prominent substrate for FAAH is not mentioned at all in this review. In contrast to anandamide (AEA) OEA is supposed to signal satiety and antagonize effects of anandamide. PEA is also not mentioned. AEA, OEA and PEA are all metabolized via FAAH and have in part opposing functions on regulation of hunger and satiety. Considering FAAH as a putative target needs to consider OEA and PEA in addition to AEA. The substrate diversity makes it obvious from the beginning that FAAH is likely not a target for obesity and would provide the molecular basis for the conclusion.
Table 1 shows only 5 studies. Are there only five pre-clinical drug-studies? what about other preclinical studies using FAAH antagonists which provide body weights of the animals?
Overall, one would like to see all 45 papers found to be relevant to be represented in one of the tables. Hence, all 45 are supposed to be revealed
Figure 1 does not tell if human and/or animal studies were included.
one would expect separate searches for genetic/polymorphism and for druf treatments.
What is meant with "synthesize" evidence?
The tables lack columns reporting the results on endocannabinoid concentration data as far as they were obtained. Studies with such proof-of-concept information would be more convincing than others
To understand the genetic animal models one would need a paragraph summarizing central and peripheral effects of encocannabinoids, for example 2AG in brain. Neuron (type)- specific knockout cannot be directly compared with a general knockout or mutation model.
Loss and gain-of-function polymorphisms have been discussed for FAAH in other context. Alternative effects are not considered in the review.
2-AG is differently associated with BMI in male and female subjects. Such differences are not considered in the review.
Clinical trials are currently investigating FAAH inhibitors such as PF-04457845 for e.g. pain. https://clinicaltrials.gov/ search term "FAAH". It would be important to mention such other indications and the putative occurrence of weight gain or loss as side effects of such treatments.
"to the best of our knowledge" statements and "further evidence is needed" are superfluous
Author Response
The authors review preclinical and clinical studies that have assessed effects of inhibition or genetic manipulation in rodents, or human genetic polymorphisms of endocannabinoid metabolizing enzymes in the context of obesity.
Overall, the review is well written. The literature searching procedures followed standard procedures. It is unclear if pre-clinical and clinical papers were searched together or separately. One would expect to use different criteria of the quality and results should be presented separately.
We thank the reviewer for their support of our manuscript. Preclinical and clinical papers were searched together as noted in the Search Strategy section. However, to make this more clear, we added a sentence on p4 that reads, “A single search was conducted that included both preclinical and clinical studies.”
The tables reflect summarize the key results. Some of the 45 papers appear to be missing.
Of the 45 papers, 5 are included in Table 1, 11 are included in Table 2 (noting one paper that appeared in both Table 1 and 2), and 30 are included in Table 3. All 45 papers are represented in the tables as they appear.
Because of sex differences of endocannabinoid associations with obesity it would be important to dissect it.
A paragraph has been added to the Discussion (p19) which provides an overview of the impact of sex on cannabinoid action relevant to energy intake: “One final important consideration is the role of sex in modulating cannabinoid ac-tion. A wealth of preclinical data have established that sex significantly impacts the role of cannabinoids in multiple physiological processes, including pain, reward and addic-tion, and hunger and energy homeostasis [79, 80]. For example, it has been demon-strated in rodent models that estradiol replacement in ovariectomized females tends to reduce energy intake in response to the CB1 receptor agonist WIN 55,212-2, while tes-tosterone replacement in orchidectomized males tends to increase energy intake, which can be blocked by the CB1 receptor antagonist AM251 [80]. Unfortunately, the majority of the preclinical studies we reviewed conducted experiments in male animals only. Given that estrogens and testosterone may have divergent effects on energy intake (at least in rodents), future studies must consider sex as a biological variable in order to better understand how manipulation of the ECS may impact obesity differently in male and female bodies.”
For genetic models, substructuring for general versus cell type specific genetic manipulation would be helpful (Table 2).
A sentence was added in the footer of Table 2 to clarify: “All gene knock-outs are global knock-outs unless otherwise indicated.” In addition, we added some further discussion on p18: “The tissue-specific effects of Mgll deletion highlight the complexity of ECS control of energy intake and body weight. For example, MAGL is involved in the hydrolysis of triacylglycerols in adipose tissue [73]. Thus, future work should consider tissue-specific manipulation of MAGL on obesity-related outcomes, which may have important im-plications for therapeutic trials in humans.”
The abstract remains rather vague and one cannot learn a key message from the abstract which one would expect as a take-home message. The abstract does not reveal the controversial pro and anti-obesogenic properties of e.g. FAAH substrates (AEA, OEA, PEA) and the specific features of 2-AG in the CNS and peripheral tissues.
The abstract has been revised to provide more specific concluding remarks.
Indeed, oleoylethanolamide which is a prominent substrate for FAAH is not mentioned at all in this review. In contrast to anandamide (AEA) OEA is supposed to signal satiety and antagonize effects of anandamide. PEA is also not mentioned. AEA, OEA and PEA are all metabolized via FAAH and have in part opposing functions on regulation of hunger and satiety. Considering FAAH as a putative target needs to consider OEA and PEA in addition to AEA. The substrate diversity makes it obvious from the beginning that FAAH is likely not a target for obesity and would provide the molecular basis for the conclusion.
We added a couple sentences to the Introduction to discuss other substrates of FAAH (OEA and PEA) – p3: “There are also other substrates of FAAH that have a role in metabolism and satiety such as oleoylethanolamine (OEA) and palmitoylethanoamine (PEA). OEA is notable in particular as it has been shown to reduce food intake and suppress appetite, which is opposite to the effects of the endocannabinoids [22]. As FAAH metabolizes both pro-appetitive and anti-appetitive lipids, inhibition of FAAH would be expected to have mixed effects on appetite and body weight.”
This has also been discussed further on p18 of the Discussion: “It is important to note that there are many substrates of FAAH other than AEA (many of which do not act at CB receptors) and that the functions of FAAH substrates are in-credibly diverse. Genetic deletion or pharmacological inhibition of FAAH produces a myriad of behavioural effects in animal models, including modulation of anxiety- and depressive-like behaviours, alterations in gastrointestinal function, decreases in pain response, decreases in inflammation, and modulation of drug-seeking behaviours and withdrawal [8]. As a result, FAAH inhibitors have been investigated in preclinical models of an incredibly wide range of pathological conditions, including pain (e.g., in-flammatory, neuropathic, cancer-associated), neurological conditions (e.g., traumatic brain injury, epilepsy, movement-related disorders), gastrointestinal disorders, cardi-ovascular disorders, and psychiatric disorders (including substance use disorders) [21].”
Table 1 shows only 5 studies. Are there only five pre-clinical drug-studies? what about other preclinical studies using FAAH antagonists which provide body weights of the animals?
We identified only 5 studies that met our criteria based on our article screening process for Table 1. We have clarified this on p4: “Studies were included only when the title and/or abstract made an explicit mention of obesity, body weight, or related endpoints.” In order to capture every study that involved manipulation of FAAH, MAGL, or DAGL and had some measure of body weight, we would have needed to do full-text screening of potentially 1,160 articles that were identified in our search, which would not have been feasible.
Overall, one would like to see all 45 papers found to be relevant to be represented in one of the tables. Hence, all 45 are supposed to be revealed
As outlined above, all 45 papers are found in the three tables.
Figure 1 does not tell if human and/or animal studies were included.
We clarified this in the caption by adding the sentence: “Both preclinical and clinical studies were included in the same article screening process.”
one would expect separate searches for genetic/polymorphism and for druf treatments.
As noted previously, we conducted only a single search for this manuscript. We had initially planned to include only studies involving pharmacological manipulation, but no studies could be located in humans, so we broadened our search strategy.
What is meant with "synthesize" evidence?
In this context, by “synthesize” we mean to “combine (a number of things) into a coherent whole” (definition from Oxford Languages).
The tables lack columns reporting the results on endocannabinoid concentration data as far as they were obtained. Studies with such proof-of-concept information would be more convincing than others
We appreciate the reviewer’s interest in seeing endocannabinoid concentration data. However, as the reviewer has raised elsewhere and as we have expanded on in the manuscript (see p18 of the Discussion), there are many substrates of FAAH, MAGL, and DAGL beyond the canonical endocannabinoids. As we specified we were interested in obesity-related outcomes, we feel that adding endocannabinoid data to the tables would be beyond the scope of our review.
To understand the genetic animal models one would need a paragraph summarizing central and peripheral effects of encocannabinoids, for example 2AG in brain. Neuron (type)- specific knockout cannot be directly compared with a general knockout or mutation model.
We added some further discussion on p18: “The tissue-specific effects of Mgll deletion highlight the complexity of ECS control of energy intake and body weight. For example, MAGL is involved in the hydrolysis of triacylglycerols in adipose tissue [73]. Thus, future work should consider tissue-specific manipulation of MAGL on obesity-related outcomes, which may have important im-plications for therapeutic trials in humans.”
Loss and gain-of-function polymorphisms have been discussed for FAAH in other context. Alternative effects are not considered in the review.
Further discussion of genetic and pharmacological manipulation of FAAH has been added on p18 of the Discussion: “It is important to note that there are many substrates of FAAH other than AEA (many of which do not act at CB receptors) and that the functions of FAAH substrates are in-credibly diverse. Genetic deletion or pharmacological inhibition of FAAH produces a myriad of behavioural effects in animal models, including modulation of anxiety- and depressive-like behaviours, alterations in gastrointestinal function, decreases in pain response, decreases in inflammation, and modulation of drug-seeking behaviours and withdrawal [8]. As a result, FAAH inhibitors have been investigated in preclinical models of an incredibly wide range of pathological conditions, including pain (e.g., in-flammatory, neuropathic, cancer-associated), neurological conditions (e.g., traumatic brain injury, epilepsy, movement-related disorders), gastrointestinal disorders, cardi-ovascular disorders, and psychiatric disorders (including substance use disorders) [21].”
2-AG is differently associated with BMI in male and female subjects. Such differences are not considered in the review.
In response to a previous issue raised by the reviewer, we have added a paragraph to the Discussion (p19) to draw readers’ attention to the important role of sex in modulating cannabinoid action. We feel that a more detailed discussion of sex is beyond the scope of this manuscript, since there is so much relevant literature to cover that relates to sex influences on cannabinoid signaling. In addition, we could not locate any specific literature finding a sex differences in 2-AG-obesity association, and in fact found the opposite – e.g., no sex differences in the relationship between 2-AG levels and BMI, percent body fat, or visceral fat area (doi: 10.2337/db06-0812).
Clinical trials are currently investigating FAAH inhibitors such as PF-04457845 for e.g. pain. https://clinicaltrials.gov/ search term "FAAH". It would be important to mention such other indications and the putative occurrence of weight gain or loss as side effects of such treatments.
We agree this is an important point we overlooked in our initial draft. We have now added a paragraph on human trials of FAAH inhibitors (and future trials of MAGL and dual FAAH/MAGL inibitors) in the Discussion on p19: “Following up on the promising findings of preclinical animal studies of FAAH in-hibition in multiple disease models, a number of human trials have been conducted or are ongoing to evaluate the potential of FAAH inhibitors across a range of indications [21]. For example, recently completed trials have found a positive signal for the FAAH inhibitor JNJ-42165279 in reducing anxiety symptoms in patients with social anxiety disorder [77] and the FAAH inhibitor PF-04457845 in reducing cannabis withdrawal in men with cannabis dependence [78]. A search of ClinicalTrials.gov identified a number of registered ongoing trials of FAAH inhibitors for substance use disorders, Tourette Syndrome, and pain. There have also been MAGL and dual FAAH/MAGL inhibitors developed to be evaluated in controlled trials. While our review has not provided clear evidence that FAAH or MAGL inhibitors would be useful for treating obesity, it will be worth noting whether there are any weight- or obesity-related effects observed in FAAH and/or MAGL trials.”
"to the best of our knowledge" statements and "further evidence is needed" are superfluous
We removed “to the best of our knowledge” from the sentence in question in the Introduction. We have also reworked other sentences to avoid using the phrase “further evidence is needed”.
Reviewer 2 Report
Matheson et al. provided an interesting review with the focus of endocannabinoid system in the treatment of obesity. The study is relative comprehensive and informative. The content is substantial, of clear logical tiers and relatively fluently expressed. A figure showing the association of FAAH, MAGL, DAGL with ECS is required.
Author Response
Matheson et al. provided an interesting review with the focus of endocannabinoid system in the treatment of obesity. The study is relative comprehensive and informative. The content is substantial, of clear logical tiers and relatively fluently expressed. A figure showing the association of FAAH, MAGL, DAGL with ECS is required.
We thank the reviewer for their support of our manuscript. We have added a figure (Figure 1) to illustrate the relationship between the enzymes and primary endocannabinoid substrates that are the focus of the review.
Round 2
Reviewer 1 Report
The authors have adequately addressed my comments.